# Lipid-Polymeric Films: Composition, Production and Applications in Wound Healing and Skin Repair

**DOI:** 10.3390/pharmaceutics13081199

**Published:** 2021-08-04

**Authors:** Eliana B. Souto, Cristiana M. P. Yoshida, Gislaine R. Leonardi, Amanda Cano, Elena Sanchez-Lopez, Aleksandra Zielinska, César Viseras, Patricia Severino, Classius F. da Silva, Raquel de M. Barbosa

**Affiliations:** 1CEB—Centre of Biological Engineering, University of Minho, Campus de Gualtar, 4710-057 Braga, Portugal; aleksandra.zielinska@igcz.poznan.pl; 2Department of Pharmaceutical Technology, Faculty of Pharmacy, University of Coimbra, Pólo das Ciências da Saúde, Azinhaga de Santa Comba, 3000-548 Coimbra, Portugal; 3Faculty of Pharmaceutical Sciences, Federal University of São Paulo, Rua São Nicolau, 210, Diadema, São Paulo 09913-030, Brazil; cristiana.yoshida@unifesp.br; 4Faculty of Pharmaceutical Sciences, State University of Campinas, Rua Cândido Portinari, 200—Cidade Universitária, Campinas 13083-871, Brazil; gislaine.leonardi@fcf.unicamp.br; 5Department of Pharmacy, Pharmaceutical Technology and Physical Chemistry, Faculty of Pharmacy and Food Sciences, University of Barcelona, 08028 Barcelona, Spain; acanofernandez@ub.edu (A.C.); esanchezlopez@ub.edu (E.S.-L.); 6Institute of Nanoscience and Nanotechnology (IN2UB), University of Barcelona, 08028 Barcelona, Spain; 7Institute of Human Genetics, Polish Academy of Sciences, Strzeszyńska 32, 60-479 Poznań, Poland; 8Department of Pharmacy and Pharmaceutical Technology, School of Pharmacy, University of Granada, Campus of Cartuja s/n, 18071 Granada, Spain; cviseras@ugr.es; 9Andalusian Institute of Earth Sciences, CSIC-University of Granada, Av. de Las Palmeras 4, 18100 Armilla, Spain; 10Institute of Technology and Research (ITP), Av. Murilo Dantas, 300, Aracaju 49010-390, Brazil; patricia_severino@itp.org.br; 11Industrial Biotechnology Program, University of Tiradentes (UNIT), Av. Murilo Dantas 300, Aracaju 49032-490, Brazil; 12Laboratory of Drug Development, Department of Pharmacy, School of Pharmacy, Federal University of Rio Grande do Norte, Natal 59078-970, Brazil

**Keywords:** biopolymers, lipid-polymeric films, lipids, skin repair

## Abstract

The use of lipids in the composition of polymeric-based films for topical administration of bioactive ingredients is a recent research topic; while few products are commercially available, films containing lipids represent a strategic area for the development of new products. Some lipids are usually used in polymeric-based film formulations due to their plasticizing action, with a view to improving the mechanical properties of these films. On the other hand, many lipids have healing, antimicrobial, anti-inflammatory, anti-aging properties, among others, that make them even more interesting for application in the medical-pharmaceutical field. This manuscript discusses the production methods of these films both on a laboratory and at industrial scales, the properties of the developed biopolymers, and their advantages for the development of dermatologic and cosmetic products.

## 1. Introduction

The skin is the largest organ in the human body, occupying an average area of 2 m^2^, which corresponds to about 10 to 15% of the total body weight [1]. It is a complex and heterogeneous covering organ [2], whose main role is to protect the organism by preventing the entry of harmful substances [3], and also by avoiding the excessive evaporation of water, which would lead to dehydration, thereby promoting a barrier function, being the epidermis the most important layer for this function [4,5,6].

The epidermis, which is represented in Figure 1, is the layer above the dermis consisting of stratified squamous epithelium. In thick skin (palms and soles), five layers can be distinguished in the epidermis. Starting from the deepest towards the surface, there is the basal stratum, the spinous stratum, the granular stratum, lucid stratum, and the stratum corneum. The lucid stratum is not present in thin skin. The cells of the epidermis are a dynamic system, i.e., they are constantly renewed, from their junction with the dermis to the skin surface, where a permanent peeling occurs [7,8]. The epidermis has four types of cells, namely Langerhans cells, keratinocytes, Merkel cells, and melanocytes, illustrated in Figure 1.

## 2. Lipid Barriers of the Epidermis

The high lipid content in the stratum corneum forms a barrier against water retention. This lipid barrier consists predominantly of ceramides, free fatty acids, and cholesterol. The presence of endogenous lipids in the skin is essential since it helps to maintain an adequate level of water in the stratum corneum, which allows skin flexibility [9,10].

Aging and environmental damage result in a reduction in the capacity of the stratum corneum to retain its ideal moisture content, making the skin dry and wrinkled [11]. Certainly, aging is a natural and inevitable process, but the incorporation of emollient components in topical formulations can be effective in preventing dehydration and in maintaining skin health. Vegetable oils have emollient properties and are therefore frequently used in dermatological and cosmetic formulations [12].

Ceramides are the most abundant lipids in the lipid barrier present in the stratum corneum of the epidermis, which has the ability to retain water in the skin. Ceramides are rich in linoleic acid (C18:2), important to maintain the barrier function of the epidermis, as it regulates trans-epidermal water loss [13]. The presence of water in the cells of the stratum corneum is responsible for the softness and elasticity of young and healthy skin. In the development of cosmetic and dermatological formulations, moisturizing and emollient actives are often applied in order to prevent the reduction of the skin’s water content [14,15]. Due to the important barrier function of endogenous ceramides to maintain an adequate skin hydration level, cosmetic formulations that have this type of lipids in their composition have often been considered potent moisturizers [13,16]. However, further studies are still needed to assess the real benefit provided to the skin by the use of exogenous ceramide.

### 2.1. Sebaceous Lipids

In addition to the lamellar lipids located in the stratum corneum and derived from lipid synthesis that occurs in keratinocytes during cell renewal, there is also a natural emulsion on the surface of human skin, which is a superficial hydrolipidic film [17]. This film is formed by mixing the sweat secretion with the lipids from the sebaceous glands. Its main function lies in protecting the skin from external aggressions, such as sun and wind. The most well-known sebaceous lipids are triacylglycerols, fatty esters, squalene, free fatty acids, sterols, and sterol esters [18,19].

Approximately 50% of sebaceous lipids are triacylglycerols, a group widely used in cosmetic formulations. Triacylglycerols are composed of 3 fatty acids as illustrated in Figure 2, attached to a glycerol molecule.

Fatty acids are long-chain hydrocarbons with carboxylated terminations. The carbon skeletons of these compounds are very stable. Fatty acids are important because they play a fundamental role in the body and are even the precursor to many cosmetic and pharmaceutical raw materials. These have hydrocarbon chains of 6 to 36 carbon atoms. Some fatty acids have a saturated chain (only with single bonds between the carbons in the chain), and others have unsaturated chains (contain double or triple bonds between the carbons in the chain). Triglycerides range between 20% and 60% in skin sebum, whereas 23–29% for wax esters, 10–14% for squalene, between 5 to 40% for free fatty acids, 1–5% for cholesterol and sterol esters, and between 1–2% for diglycerides [20].

Human skin has a number of different lipids, which are important for maintaining proper physiological conditions. In this way, the use of oils and other emollient actives in dermatological and cosmetic products can improve the spreadability of the formulation, improve sensory characteristics, promote skin hydration, and provide a pleasant sensation, presenting beneficial effects for the skin [21].

### 2.2. Lipids Used in Cosmetic and Dermatological Formulations

Many of the lipids naturally found in human skin are also used in the production of topical formulations. The knowledge of the constituents of the skin allows the development of topical biocompatible formulations. The used lipids can be of vegetable, animal, or mineral origin. They are insoluble in water and have a lower density than water. They are solubilized in organic solvents and, in general, in aqueous alkaline solutions, they are transformed into soaps [22].

However, the greatest contribution in terms of raw materials for products applied to the skin comes from the plant kingdom. Vegetable oils have fatty acids similar to those found in the human epidermis and are therefore commonly used in topical formulations.

While the composition of almond oil greatly depends on the cold press method used [23], it mainly contains 75–80% oleic acid, 10–17% linoleic acid, 5–8% palmitic acid, and 1% myristic acid, while peanut oil contains 54–76% oleic acid, 12–34% linoleic acid, 5–12% palmitic acid, 4–6% stearic acid, 3–4% arachidic acid, 2–3% behenic acid. This means that the physical properties of fatty acids and the compounds that contain them are largely determined by the length and degree of unsaturation of their chain. The non-polar hydrocarbon chain is responsible for the low solubility of fatty acids in water. The larger this chain and the smaller the number of double bonds, the lower its solubility in water. The melting points of fatty acids and the compounds that contain them are also strongly influenced by the length and degree of unsaturation of this hydrocarbon chain. At room temperature (25 °C), saturated fatty acids that have 12 to 24 carbon atoms have a waxy consistency, while unsaturated fatty acids of the same length of carbon chain are oily liquids.

The usual distinction between oils, fats, and waxes is based on their physical state: the first, liquids at room temperature, fats, soft in consistency, melt below 45 °C, the waxes are presented in solid masses, generally temperatures above 60 °C.

#### 2.2.1. Beeswax

Beeswax is a purified wax from the honeycomb of bees (*Apis mellifera*). Commercially, there are two types of waxes: yellow, natural, and white wax, obtained by discoloring the first. This whitening of the wax can be achieved chemically, by means of oxidants, or by simple exposure to light, moisture, and air (the process that gives rise to the best quality white wax) [19,24].

White wax is widely used, which, due to its high content of free fatty acids, is useful in the preparation of oil/water (O/W) emulsions. For this purpose, an alkaline product, such as sodium bicarbonate or sodium borate, is added, which will give rise to an alkaline soap. When pure, beeswax has a low water/oil (W/O) emulsifying power and can be used to increase the viscosity of formulations or to stabilize emulsions [19,24].

#### 2.2.2. Squalene

Squalene (C_30_H_50_) contains an unsaturated chain [25], it is found in shark liver oil and in some vegetable oils, and in reduced amounts in peanut and soybean oils. However, the squalene derivative (C_30_H_62_) is more frequently used in cosmetic formulations due to its greater stability when compared to squalene, thanks to its saturated carbon chain [21].

#### 2.2.3. Sterols

Usually, sterols are distinguished by their animal or vegetable origin, zoosterols, and phytosterols, respectively. Phytosterols are diverse, but the most well-known are ergostane and stigmastane, which are extracted, for example, from soy and beer yeast. The sterol characteristic of the animal kingdom is cholesterol. Its extraction is obtained from lanolin. Lanolin is a fatty material obtained from the sebaceous secretion of the sheep. It consists of a complex mixture of esters of high molar mass, sterols, and fatty acids. Lanolin consists approximately of esters, free alcohols, free fatty acids, and hydrocarbons. Among the fatty acids, the main ones are myristic and palmitic and in a smaller amount, capric and cerotic. And it can also contain water (approximately 25–30%) or be anhydrous (less than 1% moisture). The presence of sterols, such as cholesterol, gives lanolin emulsifying properties, managing to incorporate an appreciable amount of water, i.e., approximately twice its weight [21].

The biggest drawbacks of using lanolin are its unpleasant smell, the possibility of causing allergies, and the difficulty in handling, due to the fact that it has high viscosity. In this way, several lanolin derivatives have been developed in order to overcome these inconveniences and accompany the increasingly demanding and evolving cosmetic market. The lanolin derivatives that have emerged have improved organoleptic characteristics. Lanolin and its derivatives, such as lanolin alcohol, acetylated lanolin, ethoxylated lanolin are active which generally improve the spreadability of lotions and creams during application and contribute to the lubrication of the skin [21].

Other emollients commonly used in cosmetic formulations include isopropyl myristate, decyl oleate, and myristyl myristate. Isopropyl myristate (C_17_H_34_O_2_) is an ester formed by the reaction of isopropyl alcohol (C_3_H_8_O) and myristic acid (C_14_H_28_O_2_) and has an emollient action, i.e., it imparts emollience, softness, and spreading to anhydrous emulsions and preparations (Figure 3a). Decyl oleate (C_28_H_54_O_2_) is an ester formed by the reaction of decyl alcohol (C_10_H_22_O) and oleic acid (C_18_H_34_O_2_) and has an emollient action (Figure 3b). Myristyl myristate (C_28_H_56_O_2_) is an ester formed by the reaction of myristic alcohol (C_14_H_30_O) and myristic acid (C_14_H_28_O_2_), it has an emollient action and is an occlusive agent (Figure 3c). It is solid at room temperature and therefore can increase the final consistency of the product [22,24].

## 3. Films Containing Lipids

Films are flat structures formed from the deposition of a material on a surface. They can act as a semi-permeable barrier to moisture, gases, and other chemical compounds. In addition to this barrier property, films can promote hydration, remove dirt from the skin, release assets capable of promoting some aesthetic or therapeutic benefits.

The possibility of removing the film from this surface depends on both the composition of the film and the surface. Dermatological and cosmetic products administered are available on the market in two different ways: previously produced films (commonly called strips) and films formed on the skin’s own surface (covers and masks, if applied to the face). The latter form consists of an emulsion which, after spreading evenly over the skin, dries or stiffens, producing a film that can be removed from the skin surface. In this context, cosmetic masks for different purposes and depilatory waxes that can be applied to different regions of the body where there is excess hair are included. In this chapter, only the films previously produced will be covered, the strips [22].

Strips are films that have an adequate shape and size for application to specific areas of the face (Figure 4). These strips are revitalizing and provide local treatment for the areas that need them most. They are usually made of a soft polymer, which adheres smoothly to the skin and provides a continuous flow of nutrients. The Japanese company DHC (www.dhccare.com (accessed on 7 May 2021)) sells a line of strips for facial application, however, none of the products have lipid compounds in their composition, despite containing several herbal compounds such as aloe vera extract, olive leaf extract, among others.

On the other hand, the American company KAO USA Inc. owns the brand BIORÉ Skincare (www.biore.com) that produces a product called DEEP CLEANSING PORE STRIPS ULTRA. These strips are composed of polysilicon and other excipients, but the presence of two natural compounds is highlighted: oil from the leaves of Melaleuca alternifolia (Ti-tree, or Tea Tree, or Tea-tree), in addition to Hamamelis virginiana extract. According to information from the manufacturer, in just one use, these strips help to remove the dirt accumulated in the pores and even the most difficult to remove that can cause blackheads. In addition, regular use makes the pores appear smaller.

The most used materials for the production of films in the field of cosmetics and dermatology are polymers, mainly natural polymers, or biopolymers, as they have biocompatibility and reduced skin toxicity. Many biopolymers, in addition to the ability to form films, i.e., the filmogenic capacity, also have important properties for dermatological and cosmetological applications, as can be seen in Table 1. Other biopolymers such as agar, alginate, starch, and carrageenan are widely applied in the cosmetic industry as a thickener or viscosifying agent, but the literature does not report significant biological activities from the point of view of cosmetology and dermatology. Biodegradable synthetic polymers can also be used in the production of films such as poly (lactic acid), poly (lactic acid-co-glycolic acid), and polycaprolactone [26].

Natural polymers often do not produce films with adequate mechanical properties, and in order to overcome this problem, polymeric mixtures (mixtures) are often used. These mixtures can integrate natural polymers or synthetic polymers.

Additionally, collagen is a protein of fundamental importance in the constitution of the extracellular matrix of the connective tissue, being responsible for a large part of its physical properties. Its application in wound healing is desired because it stimulates and recruits immune cells and fibroblasts and promoting healing [27]. Similar properties are found in gelatin polymer.

Hyaluronic acid is a biopolymer formed by glucuronic acid and N-acetylglucosamine. It accelerates re-epithelialization and alters protein expression in the human wound model. Hyaluronic acid, a major glycosaminoglycan involved in tissue proliferation, migration, and repair, is an important factor in the activation and reepithelization of keratinocytes [28].

Chitosan is a cationic polysaccharide produced through the deacetylation of chitin, a polysaccharide found in the exoskeleton of crustaceans. Antimicrobial activity, biocompatibility, and promotes wound healing, characteristics features that make it a suitable material for wound dressings [29,30].

In addition to natural polymers, there are several synthetic polymers being used for the development of dressings, such as poly (lactic acid), poly (glycolic acid), polycaprolactone and polyethylene glycol, as they also have characteristics such as biocompatibility, and some are also biodegradable. In some cases, active ingredients, such as antimicrobials and growth factors, can be incorporated in these natural and synthetic polymers to prevent or treat infections and actively work to promote wound healing [31].

However, these polymers show inadequate mechanical properties, which can also be overcome by adding cross-linking agents to the film-forming solution. These crosslinking agents can be ionic or covalent. Ionic crosslinking agents are mandatorily used in electrostatically charged polymers, i.e., polyelectrolytes, therefore, it is an ionic interaction between polymer charges and counterion charges. On the other hand, covalent cross-linking agents are very reactive molecules that react covalently, at the same time, at two points in a polymeric chain or in different chains, promoting inter- or intra-chain bonding. The main covalent cross-linking agents include glutaraldehyde, glyceraldehyde, formaldehyde, and genipin. Some of these agents have high toxicity, such as glutaraldehyde and formaldehyde, which somewhat restricts their application. Cross-linking agents restrict the mobility of polymer chains, directly interfering with the mechanical properties of the films, however, when used in excessive quantities, the films become fragile and brittle [32].

The addition of drugs to films can induce potentially important properties for applications in cosmetology and dermatology. Drugs can be lipophilic in nature or essential oils with recognized dermatological or cosmetological properties. The application of products containing the lipids present in the skin or molecules that mimic these lipids is important to respect the physiology of human skin. The incorporation of these drugs into the polymeric film-forming solution requires a preliminary step of emulsification. It should also be noted that some lipophilic molecules, due to incompatibility with the biopolymer or due to their oxidation, may require previous solubilization in an inert lipid before the emulsification step.

The final properties of the films depend on the form of incorporation of the lipid in the film and include (i) immersion of the support (film) in a pure melted lipid solution obtaining a homogeneous layer, (ii) application on a flat support followed by drying forming an emulsified film where the lipid globules are homogeneously dispersed in an aqueous colloidal matrix, and (iii) deposit of a lipid layer in a previously formed film, which will support a bilayer or multilayer system [32].

### 3.1. Emulsification of Lipids in Polymers

Emulsions have three basic components, namely an oil phase, an aqueous phase, and a surfactant. The oily phase corresponds to the lipid that is aimed to be incorporated into the film, which is the lipid with active properties. The aqueous phase corresponds to the polymer that will form the continuous phase of the film, i.e., polymers that must be biocompatible with the skin. In this way, emulsions that are important from the point of view of film formation correspond to oil-in-water (o/w) emulsions, since the polymer will serve as the main structure of the film, i.e., the polymer will be the lipid transporter [33].

The surfactant has several functions, such as (i) the reduction of the interfacial tension between the phases, (ii) the formation of a barrier between the phases, (iii) the production of smaller droplets, and (iv) the stabilization of the dispersed phases. There are also several factors that affect the emulsification processes, such as the energy applied to the process, the emulsification time, the interfacial tension, and the viscosity of both phases [34].

### 3.2. Main Emulsification Techniques

Currently, there are several methods to prepare films to obtain emulsified films containing lipids, some of them on an industrial scale and others only on a laboratory scale. Regardless of the method employed for the production of these films, the first stage of emulsification of the lipid and (bio) polymer is necessary. It is also important to note that these emulsification techniques are precisely the same as those used for the production of emulsified creams that are marketed as cosmetic masks, i.e., films that will be formed on the skin’s own surface. Mechanical energy or pressure can be used in emulsification processes. The use of mechanical energy by means of stirrers corresponds to the most important process. As the shear promoted by the stirrer exceeds the cohesive forces of the liquid, the droplets divide into smaller sizes. The main processes that use mechanical energy are high-speed stirrers and sonicators (ultrasound probes). Common stirring processes, using mainly turbine-type stirrers, make it possible to obtain emulsions, however, the droplet size, in this case, is usually larger than the processes mentioned above. In the case of the application of pressure energy, high-pressure homogenizers are the most commonly used to produce emulsions [35,36].

### 3.3. Rotor-Stator Systems

Rotor-stator systems can be categorized into two types: (1) dispersers and (2) colloidal mills. Specifically, in the case of these high-shear homogenizers or dispersers, the rotor-stator system consists of a rotor with two or more blades and a stator with vertical or inclined openings around the homogenizing cell wall [37]. The rotor is concentrically housed inside the stator (Figure 5). As the rotor rotates, it generates a vacuum capable of moving the liquid in and out of the assembly, resulting in a circulation. The speed of rotation can reach values up to 30,000 rpm. One of the main forces that promote the reduction of the size of the dispersed droplets is the mechanical impact against the wall due to the high acceleration of the fluid. The other force is the shear force between the small gap between the rotor and the stator. At high speeds of rotation, the flow is highly turbulent and contains eddies of different scales. Obviously, the intensity of homogenization (potency) and the residence time that the droplets remain in this shear area are the main parameters that control the droplet size of the emulsion. Other parameters that can affect the performance of this type of homogenization are the viscosity of the two liquids, the surfactant, the size, volume, the ratio between the volumes of the two phases, and the geometry of the device [38].

Colloid mills are also rotor-stator systems used to obtain emulsions. The equipment consists of a pumping system that introduces the mixture of liquids that will be emulsified into a toothed stator that internally has a toothed rotor that rotates at high speed (Figure 6). The space between the stator and the rotor is very small, and the rupture of the drops in very fine droplets occurs due to the shear of these “toothed” systems. Colloid mills are commercially available on both bench- and industrial-scale [39].

#### 3.3.1. Sonicators (Ultrasound Processes)

Ultrasound consists of applying high-frequency vibrations to perform emulsification. The use of ultrasound in emulsification processes is much more efficient than the application of rotor-stator systems. Acoustic energy is mechanical energy, i.e., it is not absorbed by the molecules. Liquids irradiated with ultrasound can produce bubbles (droplets). The energy of the ultrasound is transmitted to the medium through pressure waves inducing the vibrational movement of the molecules that alternately compress and expand due to the pressure variation with time. These droplets oscillate, grow a little more during the expansion phase of the sound wave, and then shrink during the compression phase. Under the appropriate conditions, these droplets can undergo a violent collapse that generates high temperatures and pressures. This process is called acoustic cavitation [40].

Ultrasound technology is available both on a laboratory scale for processing very small volumes from 0.5 milliliters to a few liters, as well as on an industrial level that allows continuous processing of up to about 30 m^3^/h. The continuous process consists of four different parts according to Figure 7. These parts can be distinguished in the ultrasonic generator, in the ultrasonic transducer, in the flow cell, and in the tanks. In tank 1, a prior homogenization of the oil and water phases is carried out, this pre-emulsion is sent to the flow cell where the ultrasonic radiation promotes the emulsification of the two phases, which is then sent to a tank 2. Some cases require the recirculation of the emulsion from tank 2 to the flow cell to improve process efficiency.

#### 3.3.2. Membrane Emulsification

The droplet sizes of emulsions prepared with these emulsification techniques and described equipment do not produce highly monodisperse emulsions, because their emulsification conditions cannot be precisely controlled. One of the techniques that allow obtaining monodispersed emulsions is the membrane emulsification technique. In this technique, the two immiscible liquids (aqueous phase and oil phase) are separated by a membrane. The forced passage of one of the liquids through the pores of a membrane promotes the formation of droplets in the other phase, thus occurring the formation of the emulsion as can be seen in Figure 8. Porous glass membranes and the cutting diameter of these membranes are normally used as a crucial factor in the size and size distribution of the droplets obtained in the emulsion [41].

#### 3.3.3. High Pressure Homogenizers

High-pressure homogenizers are generally made up of a high-pressure pump, such as a plunger pump, which can be electrically or pneumatically driven, and a special valve called a homogenizer valve (Figure 9). The mixture of liquids (oil phase and aqueous phase) enters the valve seat and collides with a plunger pressed by a spring, which recedes a fraction of a millimeter producing a gap in which the liquid mixture can flow at an extremely high speed. When colliding with the impact ring and due to the sudden drop in pressure, the droplets are fractionated, producing the emulsion. Various constructions and geometries are available for these valves, some with or without spring. The biggest difference lies in the shape of the path and the size of the gap that must go through the mixture [42].

### 3.4. Formation of the Emulsified Film

After the formation of the polymeric emulsion and the adding of the active oil, the film can be produced by various techniques that are described below.

(a) Solvent Scattering-Evaporation Technique

The solvent spreading-evaporation technique was developed over 100 years ago driven by the needs of the emerging photographic industry. The technique consists of spreading the polymeric solution on a surface, which is then subjected to a forced air flow promoting the evaporation of the solvent and, consequently, the formation of the film on this surface. The spreading surface is located in a closed compartment, in which forced air circulation occurs in the opposite direction to the surface movement. The spreading surface can be a rotating drum or a moving belt machine (Figure 10). In the case of the rotating drum, the film is formed on the surface of a metallic drum that rotates in the opposite direction to the air flow. The metal drum has a diameter of 4 to 8 m and a width of 1.2 to 1.5 m. On the other hand, the mobile belt machine has a metal belt with a width of 1 to 2 m and a length of 10 to 100 m that circulates through two pulleys with the air flow also in the opposite direction to the movement of the belt. At the end of a turn of the drum or belt, the film is removed and proceeded to the later stages of the process [43].

On a laboratory scale, it is also possible to prepare films by the solvent spreading-evaporation technique without the need for sophisticated equipment. For this reason, this technique is widely used to establish formulations that produce films suitable for the application in question. Basically, in the laboratory, the polymeric solution (or emulsion) is spread on a flat surface, usually a petri dish. The thickness of the film is determined by the volume of solution added per square centimeter of the plate. After adding the solution, the plate is dried at room temperature or in an oven with forced air circulation. It is important that the material of the plate is compatible with the polymer and other components of the formulation, in addition, the material must provide easy removal of the film after evaporation of the solvent. Plastic petri dishes are commonly used when the polymer is water-soluble. On the other hand, glass plates are more suitable for polymers that use organic solvents [43].

(b) Hot Extrusion Technique

The main advantage of the hot extrusion technique is the absence of solvents for the solubilization of the polymer. This advantage is very attractive for polymers that require solvents that are potentially toxic for skin applications. However, the polymer must be melted to perform the extrusion [44]. A schematic of the hot extrusion technique for film production is shown in Figure 11. The technique consists of melting the polymer and extruding it in a sheet mold (flat die) to form a thin sheet of uniform and continuous thickness. The formed film is fixed on the surface of a cold molding roll (usually cooled with water) so that it is cooled as quickly as possible to a temperature below its glass transition temperature and minimizes crystallization, which would make the more fragile film. Subsequent rolls can be used to cool the film [44,45]. It is a relatively expensive technique to perform on a laboratory scale since it requires specific equipment, i.e., equipment similar to industrial equipment.

### 3.5. Electrospinning Scaffolds and Nanofibers Fabrication

Electrospinning uses electrostatic forces to produce nanofibers from polymer solutions. Basically, an electrospinning system consists of three components: a high voltage source, a syringe with capillary, and a grounded manifold. The high voltage source provides a polarity charge within the polymeric solution in the syringe, which is then accelerated to the opposite polarity collector. The polymers are firstly dissolved in selected solvents to form the polymeric solution, which is introduced into the syringe and submitted to an electric field where an electric charge is induced on its surface due to this field initially forming the Taylor Cone. When the electric field exceeds the surface tension, and a jet of the solution is ejected from the Taylor cone towards the grounded collector. While traveling through the space between the capillary tip and the collector, there is solvent evaporation, and only the polymer is collected [46].

With the expansion of this technology, several research groups have developed more sophisticated systems that can manufacture more complex nanofibers in a more controlled and efficient way. It is considered a powerful tool because it is a versatile, inexpensive, and viable technique, important for manufacturing products or structures that depend on the surface area, porosity, and surface functionality. It has been used successfully in several areas, namely, biotechnology, pharmaceuticals, production of nanocatalysts, environmental engineering, security, medicine, and especially in tissue engineering [47].

Aghamohamadi et al. (2018) fabricated an anti-bacterial fiber from Aloe vera and polyvinyl pyrrolidone, confirming the in vitro antimicrobial activity, thereby recommending it for wound healing applications [48]. Recently, Chen et al. (2021) loaded curcumin into nanofiber membranes by electrospinning technique [49]. The biomaterial showed accelerate wound healing, antimicrobial activity, and hemostatic effect in vivo, corroborating the assumption that electrospinning is indeed a promising technology for the advanced treatment of wounds.

## 4. Film Properties

Lipophilic films are generally applied onto the skin to exploit their moisture barrier properties [50]. Due to their nonpolar nature, the incorporation of a lipid into a polymeric film can make it even more lipophilic [51]. The most used lipid materials are waxes, resins, fatty acids and alcohols, acetyl glycerides, and cocoa-based compounds and their derivatives [52,53].

Lipid films have a low affinity with water molecules, depending nevertheless on the physical-chemical characteristics of the lipophilic compound [54], i.e., the polarity or the potential of the electrostatic distribution of the molecules, which depends on the chemical group, the size of the aliphatic chain and the presence of unsaturation in the chain [24].

The films obtained from natural polymers are characterized by their hydrophilic nature, which promotes their affinity with water, increasing the permeability to water vapor [55,56]. The polysaccharides generally used in the formulation of films are characterized by their selective permeability to gases, but a reduced property of water vapor barrier, due to the hydrophilic nature of these polymers [30,57,58,59,60].

Protein-based films have better barrier properties than polysaccharide-based films, due to the specific structure of proteins, which contain 20 different monomers. Polysaccharides are homopolymers and have several functional properties, mainly a high potential for intermolecular bonds, i.e., proteins can form hydrogen bonds in different positions, with different types of energy depending on temperature, solvation conditions, pH, and characteristics to incorporate other components such as plasticizers, and binding agents [61].

The addition of a lipophilic compound to the matrix of a biopolymer is carried out in order to reduce the affinity of the filmogenic matrix with the water molecules [62]. Different types of lipids can be incorporated, according to the chain size, i.e., smaller chains (fatty acids such as palmitic and lauric) and larger chains (beeswax and carnauba). Lipids can be incorporated into the formulation of polymeric films in the form of emulsified or bilayer films (the lipid is applied over the film layer). The bilayer films have better barrier properties but tend to develop deformations, such as pores, cracks, in addition to having a difficult adhesion [63]. The emulsified films are obtained through a lipid homogenization in the protein solution, forming an emulsion. These films do not have a barrier property as effective as bilayer films but are characterized by better mechanical properties [64]. Emulsified multicomponent films generally present greater opacity, being proportional to the added lipid concentration [65].

The barrier property is directly related to the chemical composition of the lipid molecules, by the presence of polar groups, size of the hydrocarbon chain, and the number of unsaturation or acetylation. Some lipophilic compounds that have been incorporated into films are listed in Table 2 [53].

The emulsified films are formed by homogenizing the lipid in the concentrated polymeric solution, followed by the drying process. Thus, a continuous polymeric matrix with small particles of lipid is obtained. The presence of these particles in the films causes an increase in the distance traveled by the water molecules during diffusion, reducing the permeability to water vapor [66]. Sherwin et al. [64] concluded that the homogenization process is an important step in obtaining emulsified films since the size of the lipid particle and its distribution are directly related to the effectiveness of their properties.

Waxes are the most efficient lipophilic compounds in reducing water vapor permeability, as they contain long chains of fatty alcohols and alkanes in their structure, which increases their lipophilicity. In order of barrier efficiency are beeswax, followed by stearic alcohol, acylglycerols, alkanes (hexatriacontane), triacylglycerols, and fatty acids (for example, stearic acid). This classification is based on lipophilicity, which defines the degree of interaction with water [53].

Stearic alcohol presents a good barrier when compared to triacylglycerols or fatty acids since the hydroxyl group has less affinity for water than the carbonyl and carboxyl groups. Considering only the polarity, the alkanes could present the best barrier to moisture, but this phenomenon is not observed, as the structure of the compound must also be considered [67].

Compounds that have the same chemical structure may have a different barrier capacity due to the size of the chain. McHugh and Krochta [68] observed that the efficiency of the moisture barrier of fatty alcohols and fatty acids increases with the number of carbon atoms (from 14 to 18), because the relative proportion of the nonpolar part of the molecule increases, reducing the solubility of water and moisture transfer.

Other authors have shown that carboxylic acids, such as stearic acid and palmitic acid, have less permeability to water vapor. These observations can be explained by the fact that long chains form heterogeneous structures in the polymeric matrix, decreasing the barrier property [69].

The chemical structure (greater polarity) and the crystalline form of unsaturated fatty acids are associated with the lower efficiency of these lipids in controlling the migration of moisture. Another factor that influences the moisture barrier is the physical state of the lipid, hydrogenated cotton oil films showed a more efficient moisture barrier, namely 300 times higher than in the liquid state, in concentrations between 0 and 40%. Paraffin increased the moisture barrier by 100 times with an increase from 75 to 100%. The structure in the solid-state is denser and limits the diffusion of water molecules, and their solubility is also reduced. However, the opposite effect may also occur, as there may be an increase in the diffusivity of water molecules, due to structural defects, such as, for example, the porosity through the surface of the lipid film [53].

Yoshida et al. (2009) [70] observed that the water vapor permeability of the emulsified chitosan films decreased with the increase in the concentration of palmitic acid in the formulation. An opposite effect was observed with the addition of beeswax and carnauba, which showed a higher moisture permeation rate than the pure chitosan film. This can be explained possibly due to the ineffective homogenization of the solutions containing the waxes, where a greater rotation would be necessary to break the chains and form a more homogeneous film. The smaller the lipid particles, the better the formation of the network, i.e., the more homogeneous the matrix is. It is important to note that the wax chains are larger than that of palmitic acid, requiring a more rigorous process of homogenization of the emulsion to break the wax chains and form a homogeneous film.

Waxes, as previously mentioned, present an excellent barrier to moisture transfer, but their efficiency cannot be associated with the polymorphism or size of the crystal. Carnauba wax promotes the formation of a porous, heterogeneous surface, while microcrystalline wax has the most uniform surface. However, the permeability of these two waxes is probably related to their affinity for water [53].

McHugh and Krochta [68] obtained films of whey proteins linked to beeswax with less permeability to water vapor. When the surface area was increased by decreasing the diameter of the lipid particles, there was a greater protein-lipid interaction. Avena-Bustillos and Krochta [71] obtained reduced values for water vapor permeability when adding beeswax to sodium caseinate films. The same result was obtained by Sapru and Labuza in films based on methylcellulose and stearic acid [72].

### 4.1. Mechanical Properties

The main mechanical properties of the films include (i) the stress at break, (ii) its flexibility, (iii) the stability to temperature changes, and (iv) the resistance in different environmental conditions and in situations with the application of physical forces [52,73].

The stress at break and the percentage of elongation represent the mechanical properties of the films, indicating the expected integrity of the film under stress conditions that could occur during processing, handling, and storage. These properties are directly related to the filmogenic structure, i.e., to the bonds present in the matrix. The stress at break is the maximum tensile force that the film can sustain before breaking, and is directly related to the order of magnitude of the energy in the intermolecular bonds present in the polymeric matrix [55,74]. The percentage of elongation corresponds to the maximum variation in the length of the sample before breaking or breaking, i.e., it measures the stretching capacity [75].

Lipid-based films are characterized by low mechanical properties. This characteristic makes it difficult to apply pure lipid films, due to their low handling and high fragility. The formation of covers on the surface can be a viable alternative of application or also the formation of emulsified films or multilayer films [76].

The addition of lipids in the biopolymer filmogenic matrix reduces the elasticity of the films, represented by the decrease in the percentage of elongation at break as a function of the lipid concentration. Moreover, the traction at the rupture of the films also proportionally reduces the addition of lipids, indicating the formation of a less rigid matrix [77].

The mechanical properties of the films depend on the interactions between the components, i.e., on the formation of strong or numerous molecular bonds between the polymer chains. Some lipids, such as acetoglycerols, fatty acids, monoacylglycerols, and phospholipids are used to increase the flexibility of polymeric films, as they present weak intermolecular forces between adjacent polymeric chains. The biggest disadvantage lies in the increased gas permeability through the film [24].

Many authors have applied proteins or polysaccharides as structural support for incorporation, and also lipids in the form of emulsified or multilayer films [58,70,77,78,79].

The addition of palmitic acid in the formulation of chitosan films reduced the permeability to water vapor in the order of 35.73% when compared to the pure chitosan film, but also promoted the reduction of the mechanical properties, in the order of 84% in the percentage of elongation at break and 77.12% of traction at break [70]. Thus, it is important to emphasize that the study of the addition of lipids is directly related to the application objective.

The microstructure evaluation of emulsified chitosan films indicated that pure chitosan films presented a cohesive, dense structure, without the presence of pores or flaws [30,80,81]. When incorporating palmitic acid, the emulsified films presented a more amorphous matrix. And with the incorporation of beeswax (0.25%, w/w), a heterogeneous matrix was formed [70,79]. The homogenization of the lipid in the polymeric solution is a very important step, as the diameter of the lipid particle and its homogeneous distribution in the matrix are directly related to the tensile at rupture and the percentage of elongation. Therefore, the smaller the diameter and the more homogeneous the distribution, the more continuous and regular the matrix of the emulsified films is [82,83]. In emulsified films, the water vapor transfer rate and the mechanical properties are dependent on the formation of the filmogenic matrix [84,85]. The films that have a more homogeneous distribution and with a smaller diameter of lipid particles are characterized by better mechanical properties and a better barrier to water vapor.

The anti-bacterial and healing properties reported for buriti oil (*Mauritia flexuosa*) [86,87], make buriti oil (*Mauritia flexuosa*) interesting for application in dermatology and cosmetics. The films developed by Batista et al. [87] were prepared using the casting technique with prior homogenization of buriti oil and chitosan solution in Ultra-Turrax at 24,000 rpm. The increase in the concentration of buriti oil promotes an increase in the number of blood cells in the structure of the film.

### 4.2. Gas Barrier Property

The barrier property of lipid films for the transfer of gases, such as oxygen and carbon dioxide, is very low when compared to films of some hydrocolloids (proteins and polysaccharides) [55], and also lower in the order of 3000 times when compared to some synthetic polymers. Many lipids, especially unsaturated ones, are subject to lipid oxidation, in the presence of oxygen, resulting in rancidity with the formation of compounds with undesirable odors [57].

## 5. Application of Films in Cosmetics and Dermatology

The preparation of a suitable vehicle is very important in the development of a dermatological and/or cosmetic formulation, as this can influence the penetration of the drug into the skin as well as its stability [88]. In addition, it can favor obtaining a product with good organoleptic characteristics, which greatly contributes to the approval and correct use of the medication by the patient [89].

Although the vehicle is, in most cases, used to deliver drugs, it can still cause some therapeutic effects, as it can provide hydration of the stratum corneum [22], being effective in the treatment of some skin conditions.

Oils and other actives with emollient capacity have often been added to dermatological and/or cosmetic vehicles in an attempt to improve the spreadability of the formulation, improve the sensory and even have some therapeutic effect [22]. In addition, as already seen, there are a number of lipids in human skin, which are important for maintaining adequate physiological conditions. This fact also favors the use of oils and their derivatives in topical formulations [13].

With technological developments and the advent of modernity, new raw materials have been introduced in the cosmetic and pharmaceutical industries. As a consequence of this evolution, there was an increase in the expectation of the effectiveness of the formulations, resulting in an increase in the number of dermatological prescriptions.

In this context, it is important for the prescribing physician to know each raw material, its effective concentrations, its incompatibilities, its possible adverse effects, and its possible interactions.

Existing patents report cosmetic products for the skin which, after application, form films with protective properties, greater elasticity, promoting the penetration of drugs into the skin, reducing the rate of water evaporation from the skin, and consequently obtaining greater skin hydration [90,91]. Specifically, hair care products contain silicones and lipids capable of forming films on the hair, providing protection and shine [92]. In this sense, nanotechnology has also contributed through the use of lipid nanoparticles that, once incorporated in cosmetic formulations, are able to form films on the skin [5,6,93,94,95,96,97,98]. The size of the lipid particles is crucial in the hydrodynamic evaporation rate of the skin’s water. The lipid nanoparticles form a layer, i.e., a film on the skin that has smaller pores when compared to the pores formed by the film produced by lipid microparticles, thus, the rate of water evaporation from the skin decreases [6].

Lulla and Malhotra filled in a patent for a transdermal pharmaceutical form capable of increasing the penetration of assets through the stratum corneum by the use of an effective, non-irritating, and safe penetration promoter [99]. The composition of this dosage form comprises a pharmaceutically active agent, an active penetration enhancing agent, a film-forming polymer, a vehicle, and optionally a propellant. The proposed product is preferably presented in the form of a spray, but it can also be a cream, lotion, gel, foam, shampoo, emulsion, or stick. Although the patent presents a wide range of actives for the composition, the authors defined as preferential the following: (i) estradiol as a pharmaceutically active agent, i.e., a hormone, (ii) egg and soy lecithins, as well as its derivatives as an active penetration enhancing agent, (iii) polyvinylpyrrolidone/vinylacetate copolymer as film-forming polymer, (iv) alcohols such as ethanol, isopropanol and propanol as vehicles, and (v) gases such as chlorofluorocarbon (CFC) or hydrofluoralkane (HFA) as propellants. The patent also proposes the incorporation of an emollient, preferably hexamethyldisiloxane. The efficacy results of this formulation are not shown. After the application of the product on the skin, a film is supposed to form on the surface that promotes the release of the active agent that penetrates the skin.

There are few studies that report the previous preparation of films for application in cosmetics and dermatology, however, the food industry has developed emulsified films with potentially important lipids for application in cosmetics and dermatology [100].

Cárdenas et al. [101] studied the medical application of chitosan films containing oleic acid or linoleic acid and glycerol. The authors tested the films on burned patients, with ulcers and injuries. The results showed that films containing glycerol showed good adhesion compared to films without glycerol. Tests on humans have shown that the films promote good epithelialization in a period of 12 to 15 days.

The scientific literature presents some works of films containing lipids and polysaccharides for application as healing dressings. Altiok et al. (2010) [102] studied the preparation of chitosan films containing thyme oil for use as a healing dressing. Thyme oil was chosen because of its anti-inflammatory, antimicrobial, and antioxidant properties. The results showed that the films show antimicrobial activity for thyme oil concentrations greater than 1.2% and the increase in oil concentration promotes an increase in the antioxidant activity of the film. These properties, added to the good vapor barrier properties of the films, make them very promising for application as a dressing, although many characterizations are necessary for cytotoxicity. Chitosan also aids in the antimicrobial properties of wound healing dressings [103,104].

The use of nano and solid lipid microparticles as delivery systems for topical, dermal, and transdermal use is an approach that, although it can be considered relatively recent, has been driving a significant increase in new formulations in the areas of dermopharmacy and cosmetics. The growing interest in the use of lipid particles for this purpose is due to its advantages over polymeric systems [105]. Among these advantages include better percutaneous absorption of drugs, biocompatibility, and less risk of toxicity, allowing its use in damaged or inflamed skin [106,107,108]. In this context, when a comparison between nano and lipid microparticles is established, it is necessary to say that the studies involving the former are considerably more extensive, which justifies a greater application in the industry. More recently, studies showing that lipid microparticles are potentially useful for the delivery of drugs both topically and transdermally by the formation of a lipid film on the skin. Among the main techniques studied include solvent evaporation, the fusion method, and high pressure cold and hot homogenization [109].

The use of the spray cooling technique for the production of lipid particles for topical use is still a minorly explored area of research [108]. Some studies have investigated the feasibility and efficiency of the production of solid lipid micro/nanoparticles through the spray colling technique for active ingredients with potential topical use [110,111,112,113].

## 6. Conclusions

Few films containing lipids are available globally in the consumer market, which makes the development of products of this nature very promising. Among the advantages of using films as a vehicle for lipid bioactive agents, their easy application, waste reduction due to dosage facilities, and the possibility to be used to disperse nanoparticulate delivery systems (e.g., liposomes, cyclodextrins, solid lipid nanoparticles, micro/nanocapsules) to modify the release profile of loaded compounds, make these biofilms very appealing, also from an economic point of view. If, on the one hand, there is an immense range of active lipid compounds already applied in cosmetics and dermatology, on the other hand, industries already have technologies for the industrial production of (bio) polymeric films.

## Figures and Tables

**Figure 1 pharmaceutics-13-01199-f001:**
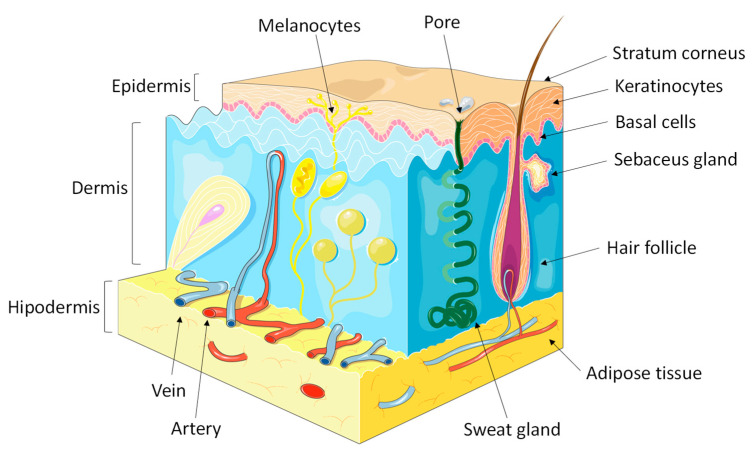
Layers, structures and cells of skin.

**Figure 2 pharmaceutics-13-01199-f002:**
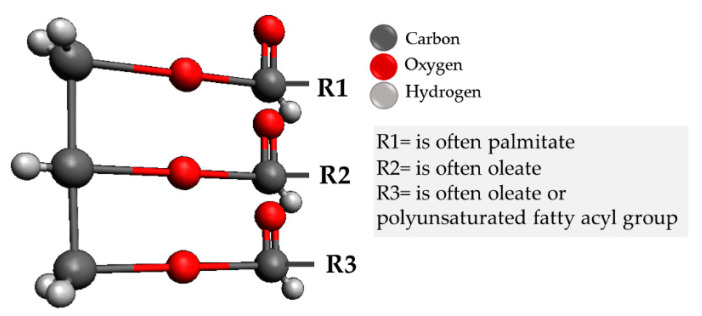
Schematic representation of the triacylglycerol molecule structure. Fatty acid is represented by R1 to R3.

**Figure 3 pharmaceutics-13-01199-f003:**
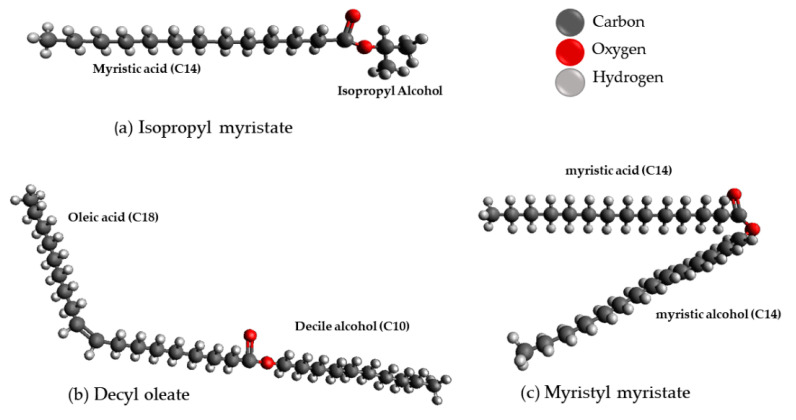
Chemical structure of emollients commonly used in cosmetic formulations, (**a**) Isopropyl myristate; (**b**) Decyl oleate and (**c**) Myristyl myristate.

**Figure 4 pharmaceutics-13-01199-f004:**
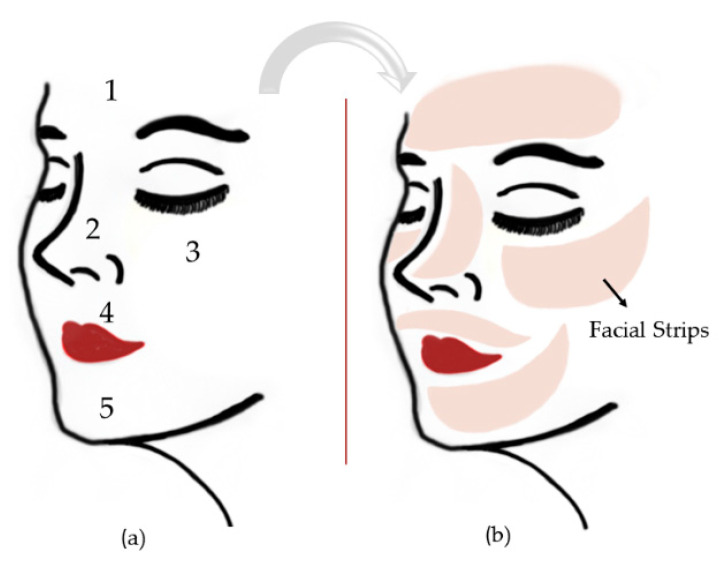
Main points to place facial strips being represented by numbers from 1–5 (**a**). Facial strips (**b**).

**Figure 5 pharmaceutics-13-01199-f005:**
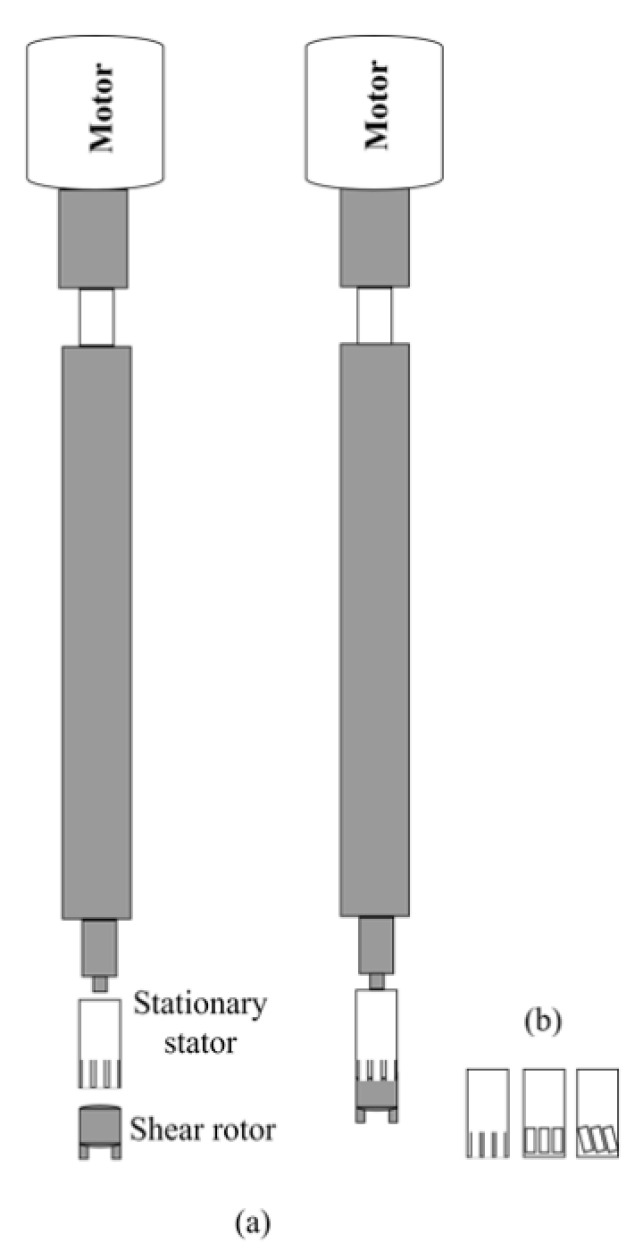
Schematic representation of the rotor-stator assembly of a high shear homogenizer type of disperser (**a**) before and after assembly; (**b**) different types of stators.

**Figure 6 pharmaceutics-13-01199-f006:**
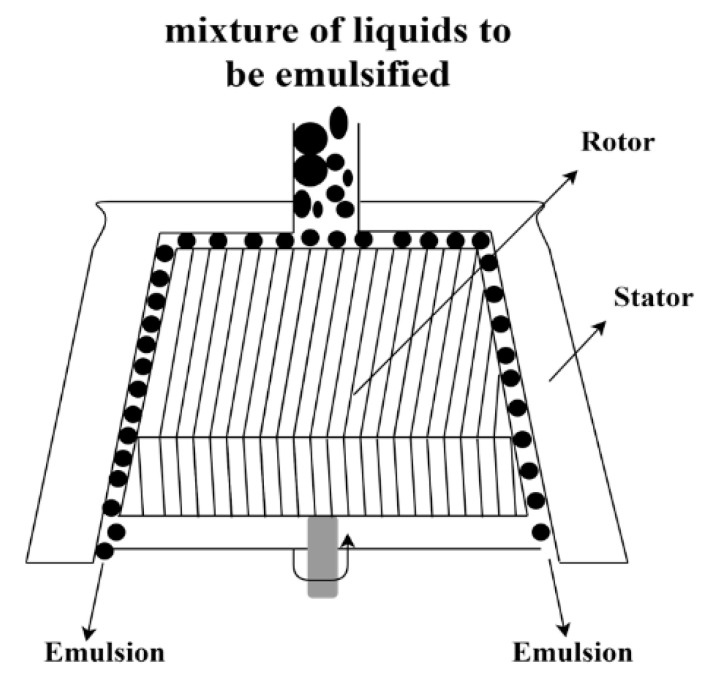
Schematic representation of a colloidal mill.

**Figure 7 pharmaceutics-13-01199-f007:**
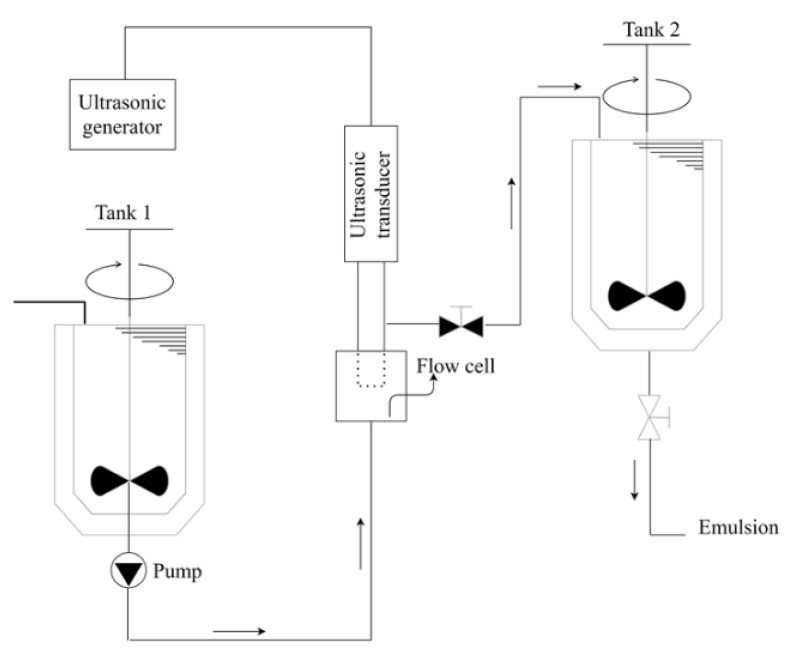
Schematic representation of the continuous emulsification system using ultrasound.

**Figure 8 pharmaceutics-13-01199-f008:**
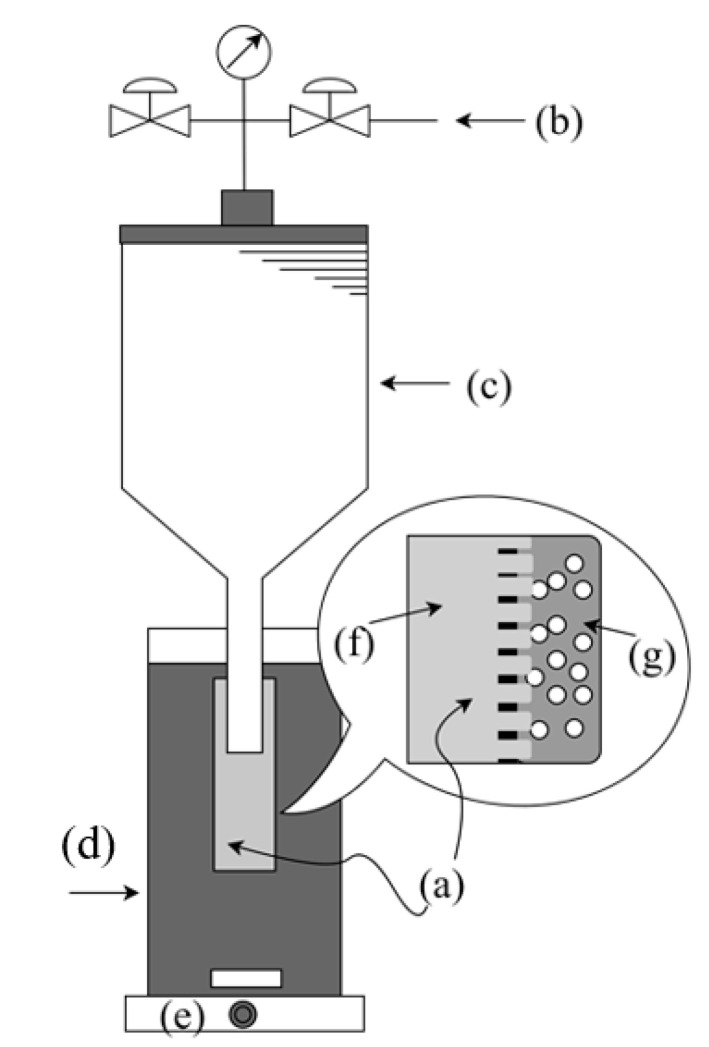
Schematic representation of the experimental system used for the preparation of O/W emulsion using porous glass membrane: (**a**) membrane; (**b**) compressed gas; (**c**) dispersed (oily) phase reservoir; (**d**) emulsion reservoir and continuous phase (aqueous phase); (**e**) and—magnetic stirrer; (**f**) dispersed phase; (**g**) continuous phase.

**Figure 9 pharmaceutics-13-01199-f009:**
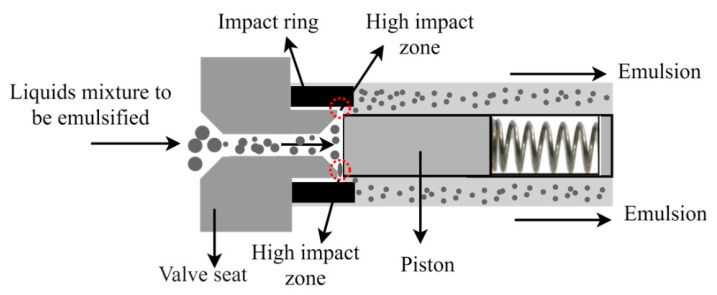
Schematic of the high-pressure valve homogenizer used to prepare emulsion with reduced size of particles.

**Figure 10 pharmaceutics-13-01199-f010:**
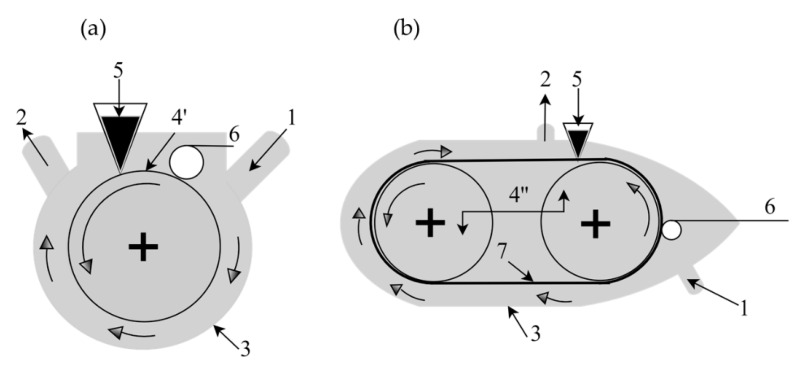
Schematic representation of the front (**a**) and cross (**b**) sections of a film production machine using the solvent spreading and evaporation technique: (1) air inlet; (2) air outlet; (3) external container; (4′) inner drum; (4′’) pulleys; and polymer spreader; (5) removed film; (6) roller; (7) moving belt.

**Figure 11 pharmaceutics-13-01199-f011:**
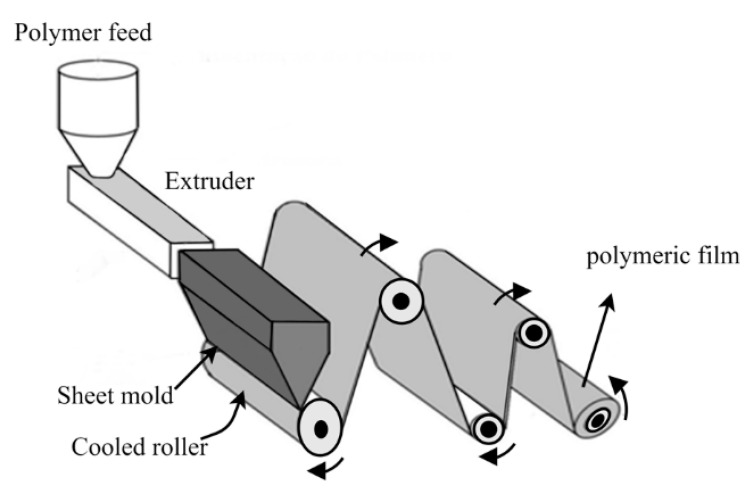
Illustration of the extrusion-spreading machine to film production.

**Table 1 pharmaceutics-13-01199-t001:** Biopolymers used in film production and their therapeutic properties.

Biopolymers	Therapeutic Properties
Fucoidan	Skin protection, antioxidant, antiaging, antiviral, anti-inflammatory, anticoagulant, and antitumor
Agar	Emollient
Collagen	Moisturizing and anti-inflammatory effect
Gelatin	Moisturizing effect and antioxidant activity
Chitosan	Healing, microbicide, mucoadhesiveness
Hyaluronic acid	Healing, swelling in soft tissue
Chondroitin and Chondroitin Sulfate	Healing and anti-inflammatory

**Table 2 pharmaceutics-13-01199-t002:** Lipophilic compounds used in films.

Lipophilic Compounds	Sources
Emulsifiers and surface-active compounds	Lecithin, mono- and di-glyceride ester, fatty acids, sucrose fatty ester
Essential oils	Essential oils of camphor, mint, and citrus
Natural resins	Chile tree (*M. zapota, M. chicle, M. staminodella, and M. bidentata*), olibanum (extracted from trees of the genus Boswellia), castor seed, and others
Oils, fats, margarine	Animal or vegetable oils and fats (peanuts, coconut, milk, sebum and others) Fractioned, concentrated and/or reconstituted oils and fats (fatty acids, mono-, di-, triacylglycerols, and others) Hydrogenated and/or trans-esterified oils (margarines, and others)
Waxes	Natural vegetable and animal waxes: carnauba, bee, whale, and others) Artificial waxes: paraffins, minerals, microcrystalline, oxidized, and non-oxidized polyethylene

## Data Availability

Not applicable.

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
