# Peer review of "Lipid-Polymeric Films: Composition, Production and Applications in Wound Healing and Skin Repair"

_pharmaceutics, 2021, doi:10.3390/pharmaceutics13081199_

Round 1

Reviewer 1 Report

This review paper is a very complete review study on lipid-polymeric films for wound healing and skin repair. The paper is well structured and written by the authors with a vast contribution from several researchers. The paper presents good insights for the new readers of the Pharmaceutics journal. The reviewer finds it suitable for its publication after attending some recommendations:

1) please, double-check the language of the paper, there are few grammar and typos mistakes in the manuscript. 

2) There are some other biomaterials (e.g. zein, Processes 20208(11), 1376) currently explored for tissue engineering (TE) and other nanotechnological application. How these lipid-based polymer films could interact with other biomaterials for TE?

3) the authors may also address the application of electrospinning as a one of the current techniques for TS applications, specially scaffolds and nanofibers fabrication.

4) The conclusions look okay for a review paper; however, it could be nice to see the future trends in the field, as well as the perspectives.  In addition to this, the authors, as experts in the field, may provide the recommendation for the new researchers in the field.

Author Response

REVIEWER #1

Comments and Suggestions for Authors

This review paper is a very complete review study on lipid-polymeric films for wound healing and skin repair. The paper is well structured and written by the authors with a vast contribution from several researchers. The paper presents good insights for the new readers of the Pharmaceutics journal. The reviewer finds it suitable for its publication after attending some recommendations:

1) please, double-check the language of the paper, there are few grammar and typos mistakes in the manuscript.

The manuscript has been thoroughly revised for its use of English, grammar, style and typos, and changes tracked for clear identification. The authors do appreciate the reviewer for the remark.

2) There are some other biomaterials (e.g. zein, Processes 2020, 8(11), 1376) currently explored for tissue engineering (TE) and other nanotechnological application. How these lipid-based polymer films could interact with other biomaterials for TE?

The authors do appreciate the reviewer’s comment on the use of other biomaterials for tissue engineering, and, in fact, there is a great deal of hybrid scaffolds that make use of lipids in their composition including for the delivery of oligonucleotides. We do hope the reviewer shares the opinion that this is a new topic beyond the scope of our manuscript. Our manuscript deals with the description of lipids used in the production of biopolymers, their production methodologies and applications. So, the main focus are the lipids. Discussing other materials and their interaction with lipid-based polymers would generate a new full-length review which is not within the aim of ours. Our work is already 26 pages long. We also don’t find how to frame the indicated reference “Processes 20208(11), 1376”, in our manuscript. We will be happy to cited, upon the suggestion of the reviewer.

3) The authors may also address the application of electrospinning as a one of the current techniques for TS applications, specially scaffolds and nanofibers fabrication.

The authors are grateful for this most valuable suggestion. We have included a new section “3.5. Electrospinning scaffolds and nanofibers fabrication” as follows: “Electrospinning uses electrostatic forces to produce nanofibers from polymer solutions. Basically, an electrospinning system consists of three components: a high voltage source, a syringe with capillary, and a grounded manifold. The high voltage source provides a polarity charge within the polymeric solution in the syringe, which is then accelerated to the opposite polarity collector. The polymers are firstly dissolved in selected solvents to form the polymeric solution, which is introduced into the syringe and submitted to an electric field where an electric charge is induced on its surface due to this field initially forming the Taylor Cone. When the electric field exceeds the surface tension, and a jet of solution is ejected from the Taylor cone towards the grounded collector. While traveling through the space between the capillary tip and the collector, there is solvent evaporation, and only the polymer is collected. With the expansion of this technology, several research groups have developed more sophisticated systems that can manufacture more complex nanofibers in a more controlled and efficient way. It is considered a powerful tool because it is a versatile, inexpensive, and viable technique, important for manufacturing products or structures that depend on the surface area, porosity, and surface functionality. It has been used successfully in several areas,  namely, biotechnology, pharmaceuticals, production of nanocatalysts, environmental engineering, security, medicine, especially in tissue engineering. Aghamohamadi et al. (2018) fabricated an anti-bacterial fiber from Aloe vera and polyvinyl pyrrolidone, confirming the in vitro antimicrobial activity, thereby recommending it for wound healing applications. Recently, Chen et al. (2021) loaded curcumin into nanofiber membranes by electrospinning technique. The biomaterial showed accelerate wound healing, antimicrobial activity and hemostatic effect in vivo, corroborating the assumption that electrospinning is indeed a promising technology for advanced treatments of wounds.”

The following references have been cited (Please see in manuscript):

  • Juncos Bombin, A.D.; Dunne, N.J.; McCarthy, H.O. Electrospinning of natural polymers for the production of nanofibres for wound healing applications. Mater Sci Eng C Mater Biol Appl 2020, 114, 110994, doi:10.1016/j.msec.2020.110994.
  • Tottoli, E.M.; Dorati, R.; Genta, I.; Chiesa, E.; Pisani, S.; Conti, B. Skin Wound Healing Process and New Emerging Technologies for Skin Wound Care and Regeneration. Pharmaceutics 2020, 12, doi:10.3390/pharmaceutics12080735.
  • Aghamohamadi, N.; Sanjani, N.S.; Majidi, R.F.; Nasrollahi, S.A. Preparation and characterization of Aloe vera acetate and electrospinning fibers as promising antibacterial properties materials. Mater Sci Eng C Mater Biol Appl 2019, 94, 445-452, doi:10.1016/j.msec.2018.09.058.
  • Chen, K.; Pan, H.; Ji, D.; Li, Y.; Duan, H.; Pan, W. Curcumin-loaded sandwich-like nanofibrous membrane prepared by electrospinning technology as wound dressing for accelerate wound healing. Materials Science and Engineering: C 2021, 127, 112245, doi:https://doi.org/10.1016/j.msec.2021.112245

Reviewer 2 Report

My report:’

Manuscript ID: pharmaceutics-1295677

Title: Lipid-polymeric films for wound healing and skin repair

Authors: Eliana B. Souto, Cristiana M. P. Yoshida, Gislaine R. Leonardi, Amanda C. Cano, Elena Sanchez-Lopez, Aleksandra Zielinska, César Viseras, Patricia Severino, Classius F. da Silva, Raquel de Melo Barbosa

Overview and general recommendation:

Overall, I found the manuscript is comprehensive work but it needs extensive revision language editing like the following (see blow). Also, the authors need to focus more on the polymers used for this purpose.

Comments to the authors:

  1. The last two sentences in the abstract need rephrasing to clarify the aim of the review.
  2. Page 2, In the epidermis structure, delete the repeated ”with melanocytes and keratinocytes”.

Author Response

REVIEWER #2

 Overview and general recommendation:

Overall, I found the manuscript is comprehensive work, but it needs extensive revision language editing like the following. Also, the authors need to focus more on the polymers used for this purpose.

The authors do appreciate very much the comments of the reviewer. The manuscript has been thoroughly revised for its use of English, grammar, style and typos, and changes tracked for clear identification.

To address the polymers used for the purpose of our discussion, we have introduced the following paragraphs: “Natural polymers often do not produce films with adequate mechanical properties, and in order to overcome this problem, polymeric mixtures (mixtures) are often used. These mixtures can integrate natural polymers or synthetic polymers. Additionally, collagen is a protein of fundamental importance in the constitution of the extracellular matrix of the connective tissue, being responsible for a large part of its physical properties. Its application in wound healing is desired because it stimulates and recruits immune cells and fibroblasts and promoting healing. Similar properties are found in gelatin polymer. Hyaluronic acid is a biopolymer formed by glucuronic acid and N-acetylglucosamine. It accelerates re-epithelialization and alters protein expression in the human wound model. Hyaluronic acid, a major glycosaminoglycan involved in tissue proliferation, migration and repair, is an important factor in the activation and reepithelization of keratinocytes. Chitosan is a cationic polysaccharide produced through the deacetylation of chitin, a polysaccharide found in the exoskeleton of crustaceans. Antimicrobial activity, biocompatibility, and promotes wound healing, characteristics features that make it a suitable material for wound dressings. In addition to natural polymers, there are several synthetic polymers being used for the development of dressings, such as poly (lactic acid), poly (glycolic acid), polycaprolactone and polyethylene glycol, as they also have characteristics such as biocompatibility, and some are also biodegradable. In some cases, active ingredients, such as antimicrobials and growth factors, can be incorporated in these natural and synthetic polymers to prevent or treat infections and actively work to promote wound healing. However, these polymers show inadequate mechanical properties, which can also be overcome by adding cross-linking agents to the film-forming solution. These crosslinking agents can be ionic or covalent.”

The following references have been cited (Please see in manuscript):

  • Kallis, P.J.; Friedman, A.J. Collagen Powder in Wound Healing. Journal of drugs in dermatology : JDD 2018, 17, 403-408.
  • Nyman, E.; Henricson, J.; Ghafouri, B.; Anderson, C.D.; Kratz, G. Hyaluronic Acid Accelerates Re-epithelialization and Alters Protein Expression in a Human Wound Model. Plast Reconstr Surg Glob Open 2019, 7, e2221-e2221, doi:10.1097/GOX.0000000000002221.
  • Matica, M.A.; Aachmann, F.L.; Tøndervik, A.; Sletta, H.; Ostafe, V. Chitosan as a Wound Dressing Starting Material: Antimicrobial Properties and Mode of Action. Int J Mol Sci 2019, 20, doi:10.3390/ijms20235889.
  • Yoshida, C.M.P.; Pacheco, M.S.; de Moraes, M.A.; Lopes, P.S.; Severino, P.; Souto, E.B.; da Silva, C.F. Effect of Chitosan and Aloe Vera Extract Concentrations on the Physicochemical Properties of Chitosan Biofilms. Polymers (Basel) 2021, 13, 1187.
  • Negut, I.; Dorcioman, G.; Grumezescu, V. Scaffolds for Wound Healing Applications. Polymers 2020, 12, 2010, doi:10.3390/polym12092010

Comments to the authors:

  1. The last two sentences in the abstract need rephrasing to clarify the aim of the review.

Abstract has been modified as requested and it is given now as follows: “The use of lipids in the composition of polymeric-based films for topical administration of bioactive ingredients is a recent research topic; while few products are commercially available, films containing lipids represent a strategic area for the development of new products. Some lipids are usually used in polymeric-based film formulations due to their plasticizing action, with a view to improve the mechanical properties of these films. On the other hand, many lipids have healing, antimicrobial, anti-inflammatory, anti-aging properties, among others, that make them even more interesting for application in the medical-pharmaceutical field. This manuscript discusses the production methods of these films both on a laboratory and at industrial scales, the properties of the developed biopolymers and their advantages for the development of dermatologic and cosmetic products.”.

  1. Page 2, In the epidermis structure, delete the repeated ”with melanocytes and keratinocytes”.

Done it.

Reviewer 3 Report

The manuscript entitled “Lipid-polymeric films for wound healing and skin repair” by Souto et al. reviews the use of pre-formed films based on lipids and polymers for dermatological and cosmetic applications.

The topic of this manuscript is interesting but some parts of this review are a very elementary description of fundamental concepts and they even contain mistakes. For instance, the sentence “Fatty acids are long-chain hydrocarbons (compounds formed by a skeleton of carbon atoms covalently linked to each other and to which only hydrogen atoms are attached) with carboxylated terminations” is an obvious statement for a researcher in the dermatological and/or cosmetic field. The authors stated that fatty acids have hydrocarbon chains of 4 to 36 carbon atoms. This statement is incorrect. Fatty acids are long-chain carboxylic acids.  Short-chain carboxylic acids (fewer than six carbons) are not generally considered as fatty acids owing to the polar nature of their acid group that makes these molecules lacking of the hydrophobic properties associated with lipids. When the carbon chain is longer than six carbons, the hydrophobic nature of the hydrocarbon acyl chain prevails over the polar acid group and the molecule shows the hydrophobic properties typical of lipids.

The meaning of the sentence “In the formulations, usually squalene is used, which has an unsaturated chain and therefore greater stability” is unclear. Why should the presence of an unsaturated chain make squalene more stable? Was squalene stability compared to what? Please, explain.

The authors describe in details the equipment and procedures used to prepare generic emulsions and films (section 3.3, 3.3.1, 3.3.2., 3.3.3., 3.4). Such descriptions are not relevant to the topic of the manuscript and they seem to be included only to lengthen the manuscript. Therefore, the above-mentioned sections should be condensed, eliminating all figures and reporting only the main concepts.

Author Response

REVIEWER #3

 Comments and Suggestions for Authors

The manuscript entitled “Lipid-polymeric films for wound healing and skin repair” by Souto et al. reviews the use of pre-formed films based on lipids and polymers for dermatological and cosmetic applications.

The topic of this manuscript is interesting but some parts of this review are a very elementary description of fundamental concepts and they even contain mistakes. For instance, the sentence: Fatty acids are long-chain hydrocarbons (compounds formed by a skeleton of carbon atoms covalently linked to each other and to which only hydrogen atoms are attached) with carboxylated terminations” is an obvious statement for a researcher in the dermatological and/or cosmetic field.

The section “(compounds formed by a skeleton of carbon atoms covalently linked to each other and to which only hydrogen atoms are attached)” has been removed from the sentence, which is now given as follows: “Fatty acids are long-chain hydrocarbons with carboxylated terminations.”.

The authors stated that fatty acids have hydrocarbon chains of 4 to 36 carbon atoms. This statement is incorrect. Fatty acids are long-chain carboxylic acids. Short-chain carboxylic acids (fewer than six carbons) are not generally considered as fatty acids owing to the polar nature of their acid group that makes these molecules lacking of the hydrophobic properties associated with lipids. When the carbon chain is longer than six carbons, the hydrophobic nature of the hydrocarbon acyl chain prevails over the polar acid group and the molecule shows the hydrophobic properties typical of lipids.

Indeed, the authors do agree with the reviewer and a new section has been introduced as follows: “These have hydrocarbon chains of 6 to 36 carbon atoms.”

 The meaning of the sentence “In the formulations, usually squalene is used, which has an unsaturated chain and therefore greater stability” is unclear. Why should the presence of an unsaturated chain make squalene more stable? Was squalene stability compared to what? Please, explain.

We have revised the text to provide the requested information, and it is now given as follows: “Squalene (C30H50) contains an unsaturated chain, it is found in shark liver oil and in some vegetable oils, and in reduced amounts in peanut and soybean oils. However, the squalene derivative (C30H62) is more frequently used in cosmetic formulations due to its greater stability when compared to squalene, thanks to its saturated carbon chain.”.

The following references have been cited (Please see in manuscript):

  • Ibrahim, N.I.; Fairus, S.; Zulfarina, M.S.; Naina Mohamed, I. The Efficacy of Squalene in Cardiovascular Disease Risk-A Systematic Review. 2020, 12, 414.
  • Leonardi, G.R.; Gaspar, L.R.; Maia Campos, P.M.B.G. Study of pH variation on the skin using cosmetic formulation s with and without vitamins A, E or ceramide: by a non-invasive method. An Bras Dermatol 2002, 77, 563-569.

The authors describe in details the equipment and procedures used to prepare generic emulsions and films (section 3.3, 3.3.1, 3.3.2., 3.3.3., 3.4). Such descriptions are not relevant to the topic of the manuscript and they seem to be included only to lengthen the manuscript. Therefore, the above-mentioned sections should be condensed, eliminating all figures and reporting only the main concepts.

The authors do appreciate the reviewer’s most valuable comment. We do hope however that the reviewer shares the opinion that production methodologies of these biofilms are part of the aim and scope of our manuscript. Sections 3.3, 3.3.1, 3.3.2., 3.3.3., 3.4 are indeed provided to cover this aim and to describe the production methods and equipment used to prepare emulsions and films, both at laboratory and industrial scale. Additionally, we showed (schematically) some of mainly the equipment used to produce these pharmaceutical forms. The authors considered this information highly relevant in its general context of this manuscript. Nevertheless, and to address this concern, abstract has been edited as follows: “The use of lipids in the composition of polymeric-based films for topical administration of bioactive ingredients is a recent research topic; while few products are commercially available, films containing lipids represent a strategic area for the development of new products. Some lipids are usually used in polymeric-based film formulations due to their plasticizing action, with a view to improve the mechanical properties of these films. On the other hand, many lipids have healing, antimicrobial, anti-inflammatory, anti-aging properties, among others, that make them even more interesting for application in the medical-pharmaceutical field. This manuscript discusses the production methods of these films both on a laboratory and at industrial scales, the properties of the developed biopolymers and their advantages for the development of dermatologic and cosmetic products.”, and title updated also to highlight the full coverage of this work as follows: “Lipid-polymeric films: composition, production and applications in wound healing and skin repair.”

Round 2

Reviewer 3 Report

The authors properly addressed most criticisms but they did not modify section 3.3, 3.3.1, 3.3.2., 3.3.3., 3.4 that they were asked to condense. As the authors are eager to describe in the details the equipment used to prepare emulsions, in section 3.3 Rotor-Stator Systems, they should include (and describe in details) the most commonly used high shear rotor/stator mixer, i.e. Silverson mixer.  A detailed description of a single equipment whose brand is clearly indicated (Ultra-Turrax™, manufactured by IKA) makes this section look like some kind of commercial for this specific equipment.

Author Response

REVIEWER #3

The authors properly addressed most criticisms but they did not modify section 3.3, 3.3.1, 3.3.2., 3.3.3., 3.4 that they were asked to condense. As the authors are eager to describe in the details the equipment used to prepare emulsions, in section 3.3 Rotor-Stator Systems, they should include (and describe in details) the most commonly used high shear rotor/stator mixer, i.e. Silverson mixer.  A detailed description of a single equipment whose brand is clearly indicated (Ultra-Turrax™, manufactured by IKA) makes this section look like some kind of commercial for this specific equipment.

Thank you very much again for the most valuable and pertinent remark with respect to section 3.3. We have revised it to remove any reference of commercial brands as, indeed, none are provided in the remaining sections of the production methods.

We reinforce the relevance of keeping the description of the production methods in detail as these are one of the main elements of the manuscript (both on a laboratory and at industrial scales), as now highlighted in the title “Lipid-polymeric films: composition, production and applications in wound healing and skin repair.”

Authors are again most thankful to the reviewer and the editor.

At disposal.

The authors.
